# Chitosan Graft Copolymers with *N*-Vinylimidazole as Promising Matrices for Immobilization of Bromelain, Ficin, and Papain

**DOI:** 10.3390/polym14112279

**Published:** 2022-06-03

**Authors:** Andrey V. Sorokin, Svetlana S. Olshannikova, Maria S. Lavlinskaya, Marina G. Holyavka, Dzhigangir A. Faizullin, Yuriy F. Zuev, Valeriy G. Artukhov

**Affiliations:** 1Macromolecules Compounds and Colloid Department, Voronezh State University, 1 Universitetskaya Square, Voronezh 394018, Russia; andrew.v.sorokin@gmail.com (A.V.S.); maria.lavlinskaya@gmail.com (M.S.L.); 2Laboratory of Bioresource Potential of Coastal Area, Institute for Advanced Studies, Sevastopol State University, 33 Studencheskaya Street, Sevastopol 299053, Russia; 3Biophysics and Biotechnology Department, Voronezh State University, 1 Universitetskaya Square, Voronezh 394018, Russia; olshannikovas@gmail.com (S.S.O.); marinaholyavka@yahoo.com (M.G.H.); artyukhov@bio.vsu.ru (V.G.A.); 4Resource Center “Molecular Structure of Matter”, Sevastopol State University, 33 Studencheskaya Street, Sevastopol 299053, Russia; 5Kazan Institute of Biochemistry and Biophysics, FRC Kazan Scientific Center of RAS, 2/31 Lobachevsky Street, Kazan 420111, Russia; dfaizullin@mail.ru; 6A. Butlerov Chemical Institute, Kazan Federal University, 18 Kremlevskaya Street, Kazan 420008, Russia

**Keywords:** graft-copolymers, chitosan, *N*-vinylimidazole, bromelain, ficin, papain, enzyme immobilization

## Abstract

This work aims to synthesize graft copolymers of chitosan and *N*-vinylimidazole (VI) with different compositions to be used as matrices for the immobilization of cysteine proteases—bromelain, ficin, and papain. The copolymers are synthesized by free radical solution copolymerization with a potassium persulfate-sodium metabisulfite blend initiator. The copolymers have a relatively high frequency of grafting and yields. All the synthesized graft copolymers are water-soluble, and their solutions are characterized by DLS and laser Doppler microelectrophoresis. The copolymers are self-assembled in aqueous solutions, and they have a cationic nature and pH-sensitivity correlating to the VI content. The FTIR data demonstrate that synthesized graft copolymers conjugate cysteine proteases. The synthesized copolymer adsorbs more enzyme macromolecules compared to non-modified chitosan with the same molecular weight. The proteolytic activity of the immobilized enzymes is increased up to 100% compared to native ones. The immobilized ficin retains up to 97% of the initial activity after a one-day incubation, the immobilized bromelain retains 69% of activity after a 3-day incubation, and the immobilized papain retains 57% of the initial activity after a 7-day incubation. Therefore, the synthesized copolymers can be used as matrices for the immobilization of bromelain, ficin, and papain.

## 1. Introduction

Biopolymers are macromolecules obtained from living organisms, and they are attracting considerable attention from researchers. The most important reasons for this are their renewability, low toxicity, and biocompatibility. Traditionally, such materials are used in the food, cosmetic, and pharmaceutical industries [1,2,3]. However, with the recent development of biomedical technologies, the greatest attention is paid to the natural macromolecules in this field [4,5,6]. For example, they are used as components in drug delivery systems [7,8] or wound dressing [9,10,11,12].

One of the promising biomaterials for medicine is chitosan [13,14]. This natural polysaccharide exhibits the properties of a polycation, which is quite rare for such polymers, and it can thus interact with a wide range of biologically active substances [15]. For example, chitosan has successfully proven itself as a matrix for the immobilization of hydrolytic enzymes [16,17]. By adsorption, it is possible to obtain immobilized papain or ficin characterized by high stability, antibacterial activity, and the ability to destruct microorganism biofilms [18,19], increasing the effectiveness of antibacterial substances.

On the other hand, chitosan is characterized by a limited pH range of solubility (p*Ka*~6.5) [20]. Due to this fact, it is difficult to obtain substances containing a sufficient amount of the immobilized enzyme. In addition, the range of chitosan solubility may not coincide with the optima of the enzyme activity. The modification of chitosan by tuning its properties can solve these problems. One of the promising methods for its modification is the graft copolymerization of chitosan with vinyl monomers [21]. For example, the *N*-vinylimidazole homopolymer, poly(*N*-vinylimidazole), PVI, is water-soluble and this will expand the range of solubility of the polysaccharide [22]. In addition, the high complexing properties of PVI to various compounds are well known [23,24,25], as well as the ability to form protein-like (co)polymers [26]. Based on these facts, macromolecules with a combination of a backbone of non-toxic biocompatible chitosan with side grafted chains from PVI will provide high efficiency for enzyme binding, forming a conjugate with low toxicity. However, it is well known that immobilized forms of enzymes are usually characterized by lower catalytic activity compared to native enzymes [16]. Therefore, when choosing a matrix for immobilizing an enzyme, one should consider not only the stability of the resulting enzyme preparation but also its catalytic activity.

Plant cysteine proteases are of particular interest for the production of enzyme formulations with a matrix based on PVI polymers. However, like many enzymes, these proteins are characterized by a low stability in aqueous solutions, which can be increased by creating immobilized forms of enzymes. The active site of cysteine proteases consisting of cysteine, aspartic acid, and histidine residues [27] contains an azole ring. Therefore, we can expect the efficient formation of a conjugate enzyme: a graft copolymer of chitosan with *N*-vinylimidazole, which is of interest for the evaluation the effect of this interaction on the preservation of the proteolytic activity of the obtained substances. In addition, cysteine proteases have found wide applications in food technologies, pharmaceuticals, and biomedicine [27]. This modification enhances the applicable value of researching immobilized enzymes.

In this regard, this work was targeted to synthesize graft copolymers of chitosan and *N*-vinylimidazole of various compositions and to study the possibility of their use as matrices for the immobilization of cysteine proteases—bromelain, ficin, and papain.

## 2. Materials and Methods

### 2.1. Materials

Chitosan (Cht) with Mw 350 kDa and 0.85 degree of deacetylation, BioProgress, Russia, and *N*-vinylimidazole (VI), Sigma-Aldrich, Germany, were used in this work. The monomer was distillated in vacuum (*T_b_* = 78–79 °C/11 mm Hg; *n*^20^*_D_* 1.5338). Potassium persulfate (PPS) and sodium metabisulfite (SMB), Vekton, Russia, were applied as initiators and were recrystallized from water before the work. DMF, acetone, ethanol, 2% *w*/*v* acetic acid, Sigma Aldrich, Germany, and distilled water were used as solvents. The cysteine proteases—bromelain (B4882), papain (P4762), and ficin (F4165)—and azocasein applied as a hydrolysis substrate were purchased from Sigma-Aldrich, Germany, and used as received without any treatment.

### 2.2. Synthesis of the Cht-g-PVI Copolymers

For the typical experiment, 0.5 g of Cht and 50 cm^3^ of 2% *w*/*v* acetic acid were placed in a thermostatically controlled reactor equipped with a mechanical stirrer. The mixture was kept at 40 ± 2 °C with stirring until the polymer dissolved completely. Then the calculated amount of PPS and SMB mixture (1:1 mol) was added to obtain initiator concentration 4 × 10^−3^ mol × L^−1^, and the calculated amount of VI aqueous solution containing HCl (1:0.1 mol to prevent PVI chain transfer reaction) after being maintained for 15 min was also appended (Table 1). After 24 h the reaction mixture was placed in a beaker containing 200 cm^3^ of acetone and was stirred. The forming precipitate was filtered off using a Buchner funnel and washed. The resulting copolymers were purified by dialysis for 1 week against distilled water using dialysis tubing cellulose membrane, cutoff 12 kDa, and freeze dried to a constant weight.

### 2.3. Instrumental Section

The IR spectra were recorded on an IR-Affinity 1 FTIR spectrometer (Shimadzu Instruments, Japan) in the ATR mode with a ZnSe prism in a frequency range from 700–4000 cm^−1^ and a resolution of 4 cm^−1^. Analyzed samples were in the form of dry powders. The composition of copolymers was calculated by FTIR data considering the ratio of the areas of the absorption bands related to the vibrations of 1,4-glycosidic bonds of Cht and C-N bonds of imidazole ring at 1189 cm^−1^ and 915 cm^−1^, respectively.

The grafting efficiency (*GE*) was calculated by the following equation:(1)$$GE=\frac{m_{Cht-g-PVI}-m_{Cht}}{m_{VI}},$$
where *m_Cht-g-PVI_*, *m_Cht_*, and *m_VI_* are the mass graft copolymer obtained, chitosan, and VI used in the polymerization, g, respectively.

The frequency of grafting (*FG*) is expressed as the number of grafted polymer chains per anhydrous glucosamine unit (*AGU*) of the backbone polymer and is obtained from the relationship [21]:(2)$$FG=\frac{PVI,\;\%}{M_{PVI}}\times \frac{M_{AGU}}{Cht,\;\%},$$
where *PVI*, % is the percentage of grafted *PVI* in Cht-g-PVI; *M_PVI_* is the molecular weight of *PVI*, *M_AGU_* is the average weight of anhydrous glucose unit of *Cht*; *Cht*,% is the percent of *Cht* in Cht-g-PVI.

To determine molecular weight of the grafted PVI chains, synthesized copolymers were subjected to oxidative degradation [22] followed by viscosimetry in ethanol at 20 °C of the isolated grafted chains. The molecular weights were calculated from the viscosity data processed by the Mark–Kuhn–Houwink–Sakurada equation [28]:


[η] = 0.0485M_η_^0.63^
(3)

The electrokinetic potential (ξ-potential) of polymer particles in aqueous solutions was determined by laser capillary Doppler microelectrophoresis with a Malvern ZetaSizer Nano instrument (Malvern Instruments, UK) in cuvettes equipped with a gold electrode at 25 °C.

Hydrodynamic diameters *D_h_* of the polymer particles in the water solution were determined by DLS with Malvern Zetasizer Nano instrument (Malvern Instruments, UK). A He-Ne laser with λ = 633 nm was utilized as a light source, the scattering angel was 170°.

Transmission electron microscopy (TEM) was performed to confirm the DLS data by using a Libra 120 Carl Zeiss electron microscope. A droplet of the sample liquid was deposited on a Formvar-coated copper grid and air-dried for 1 min, and then the excess of the solution was blotted off.

### 2.4. Enzyme’s Immobilization

The immobilization of bromelain, papain, and ficin on synthesized copolymers was performed using the adsorption approach developed previously [16]. The protein content in immobilized formulations was determined by the Lowry method [29]. Measurement of the proteolytic activity of the enzyme was carried out on the substrate azocasein [30]. The statistical significance of differences between the control and experimental values was determined by Student’s *t*-test (*p* < 0.05) since all results were characterized by a normal distribution.

### 2.5. Molecular Docking

The enzyme structures were prepared for docking according to the standard scheme for Autodock Vina package: the atoms (together with their coordinates) of the solvent, buffer, and ligand molecules were removed from the PDB input file. Before carrying out the numerical calculations, a charge was placed on the surface of proteins using MGLTools. The center of the molecule and the parameters of the box («cells») were set manually, ensuring that the whole protease molecule could fit in the box.

The Cht-*g*-PVI copolymer structure model was drawn in the HyperChem molecular designer; this structure was consistently optimized first in the AMBER force field, and then quantum-chemically in PM3. The ligands in the docking calculations had the maximum conformational freedom; rotation of the functional groups around all single bonds was allowed. Charge arrangement on the Cht-*g*-PVI copolymer molecule and its protonation/deprotonation was performed automatically in the MGLTools 1.5.6 package (https://ccsb.scripps.edu/mgltools/1-5-6/, accessed on 15 May 2022) [16].

## 3. Results and Discussions

### 3.1. Synthesis and Characterizations of the Cht-g-PVI Copolymers

The graft copolymers of chitosan and *N*-vinylimidazole with different compositions denoted as Cht-*g*-PVI were synthesized via free radical solution polymerization in the presence of a PPS-SMB system. The temperature and pH of the reaction medium have a significant effect on the polymerization process and architecture of the resulting polysaccharide-based polymers. According to the research data [31], low pH values and high temperature during the polymerization lead to the formation of carbohydrate block copolymers, while mild conditions favor the formation of graft copolymers with a polysaccharide backbone and synthetic side chains. Moreover, heating a highly acidic solution of carbohydrates (pH~1–2) leads to an increase in the resulting product’s polydispersity due to the destruction of the initial macromolecules. Acid degradation of the polysaccharides can also lead to the accumulation of toxic degradation by-products, which can have a carcinogenic effect. For this reason, we have chosen mild conditions that contribute to the production of graft copolymers.

Another important factor determining the efficiency of the graft polymerization process is the initiator. Various types of initiation are known to be used to obtain graft copolymers. However, for solution polymerization, redox initiators are most often used, triggering a radical process. Transition metal compounds such as cerium (IV), manganese (VII), iron (II), etc., are often used in such systems [32]. However, it is well known that the most complete and efficient redox reaction occurs in an acidic medium. The undesirable degradation of polysaccharides can occur under these conditions. Another widely used initiator is potassium persulfate, which effectively initiates polymerization over a wide pH range. The operating temperature of potassium persulfate starts at 50 °C, and it is proposed to use the PPS-SMB system to reduce it. It is known that such a combination not only expands the range of PPS operation but also reduces its destructive effect on polysaccharide macromolecules. That is why we used the initiating system PPS-SMB in a molar ratio of 1:1 in this work.

It is well known that VI tends to participate in the side chain transfer reaction due to the formation of macroradical resonance forms (Figure 1) [33]:

To prevent this, a quantity of HCl (VI:HCl = 1:0.1 mol) was added. The pH of the resulting polymerization mixture was in the range of 3.9–4.5 which allows it to achieve mild polymerization conditions resulting in a graft polymer.

The polymerization mechanism includes the following steps. Firstly, PPS and SMB dissociate in an aqueous medium, and metabisulfite ions interact with water molecules turning into hydrosulfite anions. Persulfate anions react with HSO_3_^−^ ions forming sulfate ion-radicals and hydrosulfite radicals which interact with the water molecules resulting in hydroxide radicals [34]. Then, all the types of the formed radicals react with the OH or NH_2_ groups of the chitosan generating macromolecule radicals. These radicals interact with *N*-vinylimidazole molecules forming growing polymeric chains. The chain terminations proceed due to the recombination. The polymerization scheme is represented below (Figure 2).

The structure of the synthesized copolymers was confirmed by FTIR (Figure 1). The FTIR spectrum contains the following characteristic absorption bands: at 915 cm^−1^ attributed to the deformation vibrations of the imidazole cycles; at 1057 and 1098 cm^−1^ corresponding to the stretching symmetric skeletal vibrations of the pyranose cycles and the C-O-C bond, respectively; at 1189 cm^−1^, due to 1,4-glycosidic bonds; at 1228 cm^−1^ relating to the stretching vibrations of the C-N bonds and the deformation vibrations of the C-H bonds of the imidazole cycles; at 1588 cm^−1^, describing the deformation vibrations of the N-H bonds of the chitosan amino groups; a band at 2875 cm^−1^ ascribing to the stretching vibrations of the methylene groups; and a wide band in the region of 3000–3500 cm^−1^ related to vibrations of OH and NH_2_ groups of chitosan [35]. Several of the bands at 1320, 1412, and 1500 cm^−1^ corresponding to the vibrations of the dissociated carboxylic groups indicate that the obtained copolymers were in the acetate salt form. The absence of the low-intensity absorption band near 1610 cm^−1^ confirms that graft polymerization occurs due to the opening of the C=C bond of *N*-vinylimidazole [35,36].

The graft copolymer compositions were also determined by FTIR data and are presented in Table 2. As it can be seen, the VI content in the copolymers grows with an increase in the VI content of the polymerization mixture. In addition, the same pattern is observed for the molecular weights of the PVI grafted chains, as these values also rise with VI content growth in the reaction feed. However, the graft–copolymer yields decrease with the VI content growth in the reaction feed. This is due to the enhancing side processes such as VI homopolymerization. The yield data correlates to the grafting efficiency (GE) values which also decrease due to side processes occurring.

The frequency of grafting (FG) that was obtained was relatively high despite the chitosan’s molecular weight. It is shown that chitosan functional groups were available for interactions with radicals and monomer due to the chitosan concertation in the polymerization solution being lower than the critical coil overlap concentration [22].

So, the acquirement of the chitosan and *N*-vinylimidazole graft copolymers with different compositions via radical solution polymerization was confirmed by FTIR. It was found that graft copolymer yields and grafting efficiency decrease with VI content growth in the polymerization feed. On the other hand, the molecular weights of the grafted PVI chains and the frequency of the grafting increases with a monomer content rise. Moreover, all the synthesized graft copolymers are characterized by the high frequency of PVI grafting.

### 3.2. Copolymer Solution Properties

It is well known that chitosan has been widely used in biomedicine due to its beneficial properties such as biocompatibility, low toxicity, and film-forming ability, and it is soluble in aqueous solutions with pH < 6.5 by amino group protonation turning into ionized polycation which is a rarity for biopolymers. The grafting of some side water-soluble polymeric chains can enhance the pH solubility range keeping the chitosan’s unique cationic behavior. Poly(*N*-vinylimidazole) is a water-soluble synthetic polymer with low-basic properties (p*Ka*~6) and high complex ability for a variety of different substances [22,23,24,25]. Cht-*g*-PVI copolymers are expected to retain their polycationic behavior and be pH sensitive in solutions. So, the next part of our investigation deals with the research of the copolymer solution properties.

All the synthesized copolymers are soluble in distilled water (pH = 6.7 ± 0.02) and their 1% *w*/*v* aqueous solutions can be obtained. These are the polymer strength solutions, and they are opalescent viscous liquids. Thus, 0.1% *w*/*v* solutions were chosen for further research.

The relatively high viscosity of the obtained solutions can indicate the copolymer macromolecule’s self-association. To research the copolymer association behavior in the solution the DLS method was used. The results of the hydrodynamic diameter (*D_h_*) measurements are presented in Table 3. The data obtained show that copolymer macrochains are aggregated, and *D_h_* values significantly increase with the growth of the PVI content in the copolymers. Therefore, PVI links act as a trigger for the self-association of the macromolecules. These results correlate with the earlier obtained data for azole-containing copolymers [36], and the DLS size data are in good agreement with the size results obtained by TEM (Figure 2). The copolymer particles have an oval-like shape and are characterized by a dense-core structure with less core corona. The self-associated polymers are characterized by critical coil overlap concentration, *c**, which was determined by DLS (Figure 3A). The solutions were researched in the concentration range of 10*^−^*^3^–0.1% *w*/*v* and a sharp increase in the size of the copolymer particles was taken as *c** (Table 3). As it can be seen, *c** values increase with PVI copolymer content and PVI grafted chain molecular weight. These results are in good agreement with the earlier obtained data [22].

To confirm the polycationic behavior of the synthesized Cht-*g*-PVI copolymers, ζ-potential measurements were carried out (Table 3). As it can be seen from the data obtained, ζ-potential positive values in the range 44–55 mV characterize Cht-*g*-PVI copolymer solutions. Moreover, ζ-potential values rise with the PVI content growth in the copolymer. The mobility and conductivity measurement data correlates with the ζ-potential indicating the contribution of PVI links to the charge of the copolymers. The high values of the ζ-potential can be also attributed to the formation of chitosan acetate from the chitosan and imidazole cycles due to the presence of acetic acid in the polymerization feed.

The presence of the polymer charge may indicate the pH sensitivity properties. Therefore, the research on the effect of pH on the ζ-potential and *D_h_* values was conducted by DLS (Figure 3B). As expected, the researched parameters increase in the acidic medium and decrease in the alkaline environment. In an acidic medium, the available imidazole rings and chitosan amino groups are protonated and the electrostatic repulsion of similarly charged fragments of the macromolecule associates is enhanced. This leads to an increase in the ζ-potential and size of the latter. In an alkaline medium, protonation does not occur, and the size and potential of the particles decrease.

In brief, synthesized Cht-*g*-PVI copolymers are water-soluble and self-assembled in the concentration range of 10*^−^*^3^–0.1*% w/v*. The critical coil overlap concentration of the synthesized copolymers rises with the increase of the PVI content and molecular weight. Cht-*g*-PVI copolymers are characterized by a cationic nature and pH sensitivity correlating with the PVI content.

### 3.3. Enzyme Immobilization and Interactions with the Cht-g-PVI Copolymer

Polymer-immobilized enzymes are widely used for various applications in biotechnology, cosmetology, and pharmacy. This is due to a biocatalyst stability increase and environmental enzyme protection. The most promising immobilization method is adsorption, i.e., a process without the covalent binding of a protein to a polymer. The use possibility of a polymer as a matrix for enzyme immobilization is determined by their ability to interact with each other. Therefore, the interaction possibility of the synthesized copolymer with the enzyme was studied by FTIR before the immobilization experiments. The Cht-*g*-PVI-3 copolymer was chosen for research because it is characterized by the lowest molecular weight of grafted side chains and the smallest particle size in solutions. This should provide greater steric availability of the copolymer functional groups for interaction with the enzymes.

The FTIR spectrum of a graft copolymer and ficin blend shows the characteristic absorption bands of the copolymer described above (Figure 1). Moreover, there is an intensity increase of the bands at 915, 1057, 1228, and 2875 cm^−1^, and shifts in the wavenumber of some bands, from 1057 to 1030 cm^−1^ (pyranose cycles) and from 1588 to 1596 cm^−1^ (chitosan amino groups). This indicates that the conjugate formation between the enzyme and the Cht-*g*-PVI-3 copolymer, as well as hydroxy-, amino groups, and imidazole cycles, are significantly involved in the conjugation process. The FTIR spectra of the immobilized papain and bromelain are the same and correlate with the results obtained for ficin.

As mentioned above, cysteine proteases are hydrolytic enzymes united by the presence of a cysteine in the active site and catalyzing the hydrolysis of proteins and peptides. The catalytic activity of cysteine proteases is due to the presence of a catalytic dyad, according to some sources, a triad, which includes the imidazole base from a histidine residue, and an activating nucleophilic center which is the thiol group of cysteine [37]. If we consider the active site of these enzymes as a triad, then it also includes the carboxyl group of aspartic acid, which shifts the electron density in the azole ring. The thiol group is a strong reducing agent, easily oxidized under fairly mild conditions. Therefore, for the industrial use of cysteine proteases, it is advisable to choose their immobilized forms, in which the active site is protected from the negative effects of the environment, and thus the catalytic activity of the enzyme is maintained for a longer time [16].

The results of the enzyme content evaluation after immobilization are represented in Table 4. As it can be seen from the data obtained, grafting of the VI links to chitosan significantly increases the amount of the bounded enzymes due to the appearance of new centers to interact with proteins. The proteolytic activity (U × mL^−1^) of immobilized enzymes was 77.5 ± 1.7 for bromelain, 39.4 ± 8 for ficin, and 96.6 ± 1.5 for papain. The values obtained are lower than those for native bromelain and ficin, however, they are the same for papain (Table 4). These results demonstrate that bromelain and ficin change the conformations to less active ones, but papain conformation is the same in the main. Moreover, the intensity increase of the absorption bands attributed to the imidazole cycles in the FTIR spectrum of the immobilized ficin indicates that the enzyme interacts with the Cht-*g*-PVI copolymer via a histidine azole ring, probably including the histidine residue from the ficin active site, and the proteolytic activity of the enzymes immobilized on chitosan are higher compared to the Cht-*g*-PVI-immobilized enzymes, which confirms the interaction of the PVI chains with proteins.

Despite some decreasing of the protease catalytic activity, the immobilized enzymes better retain their properties during this time. In the next series of experiments, we evaluated the stability of the immobilized enzymes obtained by measuring the residual catalytic activity (U × mL^−1^) after incubation at 37 °C in 50 mm Tris-HCl buffer, pH 7.5.

After a one-day incubation, the immobilized ficin retains more than 97% of its activity, while the native one retains only about 51%, and bromelain retains 84% and 53% of proteolytic activity for the immobilized and native one, respectively (Figure 4). The immobilized papain is characterized by 80% activity after a five-day incubation and by only 57% for the native enzyme. Moreover, as it can be seen, the immobilized ficin is six times more active compared to the native one and possesses 57% activity. After 7 days or more, the differences in the catalytic ability loss of the native and immobilized enzymes became more significant. The activity of the free enzymes after 21 days of incubation was 3.8% for ficin, 15% for papain, 5.8% for bromelain, and for immobilized enzymes the values are the following: 32% for ficin, 21% for papain, and 20% for bromelain (Figure 4).

For a detailed analysis of our results, we conducted in silico experiments on molecular docking. The higher the modules of the enzyme affinity values (kcal/mol) for Cht-*g*-PVI (Table 5), the greater the amount of protein sorbed onto the carrier (Table 4), but for catalytic activity the situation was much more complicated. Figure 5 and Table 5 show bonds and interactions between bromelain (PDB ID:1W0Q), ficin (PDB ID:4YYW), papain (PDB ID:9PAP), and Cht-*g*-PVI, which arise as a result of enzyme immobilization. It was established that for all three enzymes during their sorption immobilization, the polymer molecule is located near the cleft with the active site between two domains. This should reflected—and according to our data really reflected—in the catalytic activity of the samples. Ficin, which forms eight hydrogen bonds with Cht-*g*-PVI after immobilization, loses its activity to a greater extent than other enzymes. Papain, which forms two hydrogen bonds, does not reduce it at all. Moreover, in the papain molecule, hydrogen bonds are formed with the amino acids of the active site—Cys25 and His159. Obviously, its catalytically optimal conformation is stabilized. Bromelain occupies an intermediate position regarding the loss of activity during immobilization and forms six hydrogen bonds with Cht-*g*-PVI, yet not chemical bonds, but physical interactions are with the active site of the enzyme (Cys26 and His158).

Thus, it was found that synthesized Cht-*g*-PVI copolymers form conjugates with ficin, papain, and bromelain. The interaction mechanism of the synthesized copolymer and cysteine proteases was researched by FTIR and produced by molecular docking. The results of in silico research and proteolytic activity determination are in good agreement. The proteolytic activity of the immobilized enzymes is lower compared to native for bromelain, at 80%, and it is 41% for ficin, however, papain maintains full proteolytic activity. Moreover, the immobilized enzymes are more stable in solutions and retain up to 84% and 32% of initial activity after 3-day and 21-day incubations, respectively.

## 4. Conclusions

By conducting this research, we demonstrate that a potassium persulfate-sodium metabisulfite blend is a suitable initiator for obtaining chitosan and poly(*N*-vinylimidazole) graft-copolymers with a relatively high grafting frequency and efficiency. The synthesized copolymers are water-soluble despite their PVI content, demonstrating their polycationic and pH-sensitive nature, and alongside their self-assembled properties, synthesized copolymers conjugate cysteine proteases, such as ficin, papain, and bromelain. The immobilized proteins retain up to 100% of their proteolytic activity compared to the native ones, and moreover, they are more stable in solutions. The immobilized ficin retains up to 97% of its initial activity after a one-day incubation, while the native one retains only about 51%; the immobilized bromelain retains 69% of activity after a 3-day incubation, while for the native one the value is 41%. The immobilized papain retains 57% of its initial activity after a 7-day incubation, and the difference between the immobilized and native forms increases after 7 days. Therefore, the synthesized copolymers can act as promising materials for cysteine protease immobilization materials to prolonging their catalytic activity.

## Data Availability

Not applicable.

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
