# Peer review of "Chitosan Graft Copolymers with N-Vinylimidazole as Promising Matrices for Immobilization of Bromelain, Ficin, and Papain"

_polymers, 2022, doi:10.3390/polym14112279_

Round 1
Reviewer 1 Report
The paper entitled "Chitosan graft copolymers with N-vinylimidazole as promising matrices for immobilization of bromelain, ficin, and papain" by Andrey V. Sorokin and co. obtained graft copolymers of chitosan and N-vinylimidazole of various compositions and to study the possibility of their use as matrices for immobilization of cysteine proteases – bromelain, ficin and papain.
The paper is very interesting, but in my opinion some some new resuls must be added, as follows:
Some previous studies show that aggregates of polysaccharide macromolecules in aqueous solutions can have a geometry other than spherical [Kuznetsov V. A., Sorokin A. V., Lavlinskaya M. S., Sinelnikov A. A., Bykovskiy D. V. Graft copolymers of carboxymethyl cellulose with N-vinylimidazole: synthesis and application for drug delivery. Polymer Bulletin . 2019;76: 4929–4949. https://doi.org/10.1007/s00289-018-2635-0 ] as considered by the method of dynamic light scattering (DLS) to determine the size of such particles. Therefore, to study solutions of graft copolymers, it useful to determine the hydrodynamic diameters Dh from the method of transmission electron microscopy to check if the results presented in table 3 are correct
Author Response
According to your comment, the TEM data confirming particle size results were added to the manuscript. (Page 4, lines 135-138, page 6, lines 267-269, highlighted by yellow and Fig. 3)
Thanks for your comments. You help us become better!

Reviewer 2 Report
The authors describe the preparation of poly(vinylimidazole)-grafted chitosan for the immobilization of selected enzymes.
Lines 116 and 120: “anhydrous glucose unit of Cht”: the repeating unit of chitosan is D-glucosamine (assuming full deacetylation), not “anhydrous glucose”. Please correct!
Table 2: please check this entry: “FGЧ102”.
Author Response
According to you comment, the name of the chitosan repeating unit (Page 3, lines 116, highlighted by yellow), and frequency grafting abbreviation in Table 2 were corrected.
Thanks for your comments. You help us become better!

Reviewer 3 Report
I consider that the Manuscript number polymers-1751381, entitled “Chitosan graft copolymers with N-vinylimidazole as promising matrices for immobilization of bromelain, ficin, and papain” is eligible to be accepted in this journal. The description of the assays and experiments is carried out in detail, as well as the results obtained.
Author Response
Thanks for your work!
Round 2
Reviewer 1 Report
The paper can be published in prresent form